# Adaptive Optics-Transscleral Flood Illumination Imaging of Retinal Pigment Epithelium in Dry Age-Related Macular Degeneration

**DOI:** 10.3390/cells14090633

**Published:** 2025-04-24

**Authors:** Laura Kowalczuk, Rémy Dornier, Aurélie Navarro, Fanny Jeunet, Christophe Moser, Francine Behar-Cohen, Irmela Mantel

**Affiliations:** 1Faculty of Biology and Medicine, University of Lausanne, Jules-Gonin Eye Hospital, Fondation Asile des Aveugles, 1004 Lausanne, Switzerland; laura.kowalczuk@gmail.com (L.K.); irmela.mantel@svmed-hin.ch (I.M.); 2Laboratory of Applied Photonic Devices (LAPD), School of Engineering, École Polytechnique Fédérale de Lausanne (EPFL), 1015 Lausanne, Switzerland; remy.dornier@epfl.ch (R.D.); christophe.moser@epfl.ch (C.M.); 3Centre de Recherche des Cordeliers, UPMC—Paris 6, INSERM U1138, Université Sorbonne Paris Cité, 75006 Paris, France; 4Assistance Publique Hôpitaux de Paris, Ophtalmopôle, Cochin Hospital, 75014 Paris, France; 5Hôpital Foch, Service D’ophtalmologie, 92150 Suresnes, France

**Keywords:** blood–retinal barriers, retinal pigment epithelium, clinical eye research, high-resolution retinal imaging, age-related macular degeneration

## Abstract

Adaptive optics-transscleral flood illumination (AO-TFI) is a novel imaging technique with potential for detecting retinal pigment epithelium (RPE) changes in dry age-related macular degeneration (AMD). This single-center prospective study evaluated its ability to visualize pathological features in AMD. AO-TFI images were acquired using the prototype Cellularis^®^ camera over six 5 × 5° macular zones in patients with good fixation and no exudative changes. Conventional imaging modalities, including spectral-domain optical coherence tomography (OCT), color fundus photography and fundus autofluorescence, were used for comparison. AO-TFI images were correlated with OCT using a custom method (Fiji software, v. 2.9). Eleven eyes of nine patients (70 ± 8.3 years) with early (n = 5), intermediate (n = 1) and atrophic (n = 5) AMD were analyzed. AO-TFI identified relevant patterns in dry AMD. RPE cell visibility was impaired in affected eyes, but AO-TFI distinguished cuticular drusen with hyporeflective centers and bright edges, large ill-defined drusen and stage 3 subretinal drusenoid deposits as prominent hyperreflective spots. It provided superior resolution for small drusen compared to OCT and revealed crystalline structures and hyporeflective dots in atrophic regions. Atrophic borders remained isoreflective unless RPE displacement was absent, allowing precise delineation. These findings highlight AO-TFI’s potential as a sensitive imaging tool for characterizing early AMD and clinical research.

## 1. Introduction

Since the early 1990s, significant advancements have been made in retinal research, largely due to the development of advanced imaging technologies such as optical coherence tomography (OCT), fundus autofluorescence (FAF) and infrared (IR) imaging. These methods have since become the standard for diagnosing various ophthalmic diseases, most notably age-related macular degeneration (AMD), a condition associated with severe and progressive vision loss. AMD is a leading cause of blindness among adults over the age of 60 worldwide, highlighting the critical need for effective diagnostic strategies [1,2]. Global estimates indicate that AMD affects approximately 200 million people worldwide, with approximately 34 million cases reported in the European Union. Projections based on population growth and aging suggest that the number of AMD cases will increase by nearly 25% by 2050 [3,4].

Common multifactorial retinal diseases, including AMD, involve alterations in the deepest cell layers of the retina, particularly the retinal pigment epithelium (RPE). These changes ultimately lead to vision and visual acuity loss, both of which severely impact quality of life [2,5]. The exact mechanisms behind RPE/choroid and photoreceptors degeneration remain incompletely understood. One of the earliest detectable changes in AMD identified by FAF is the redistribution and loss of autofluorescent granules, primarily lipofuscin pigment. This redistribution in individual RPE cells is associated with cytoskeletal changes, including (1) loss of the sharp polygonal appearance of the cell, which become round or concave, (2) weakening of cell junctions, leading to separation, interruption and cytoskeleton disruption and (3) development of stress fibers [6,7]. Interactions between the intracellular cytoskeleton and the extracellular matrix are critical for the diverse functions of the RPE. Thus, early cytoskeleton changes may affect photoreceptors. Oxidative stress, inflammatory responses (including complement pathway activation), and lipid deregulation in the RPE, all of which play a role in AMD, contribute to cytoskeleton activation [1,8,9,10]. For example, oxidative stress has been shown to induce stress fibers and to disrupt the RPE barrier function [11,12,13] cytokines alter the RPE barrier [14] and complement pathway activation causes both direct and indirect damage to RPE cells [13,15].

While standard diagnostic imaging techniques have been effective in detecting intermediate and late-stage AMD, they are unable to detect the microscopic cellular changes associated with early stage AMD. The advent of adaptive optics (AO) has opened up new possibilities in ophthalmic research. Techniques such as AO scanning laser ophthalmoscopy (AO-SLO) and AO flood illumination (AO-FI) allow detailed imaging of individual photoreceptors, but are less efficient at visualizing individual RPE cells [16]. AO-transscleral flood illumination (AO-TFI) is an imaging modality that combines AO with oblique illumination of the retina, offering significant advantages over transpupillary illumination-based methods. This newer technique has demonstrated the potential to provide clearer visualization of individual RPE cells and detailed assessment of their microscopic morphological features [17]. AO-TFI has shown success in visualizing RPE cells more clearly, without the limitations associated with AO-SLO alone. To date, the effectiveness of AO-TFI using the Cellularis^®^ prototype camera has been evaluated in healthy eyes and in central serous chorioretinopathy patients [18,19]. However, research into the use of AO for the early detection of AMD remains limited [20,21,22,23,24,25].

Given the potential for early detection of AMD by observing cellular changes in the RPE, we sought to explore the potential of AO-TFI to detect early signs in the pathogenic pathway of AMD. Additionally, our goal was to use this new imaging modality to identify known pathologic changes in AMD patients such as drusen, subretinal drusenoid deposits and atrophy.

## 2. Materials and Methods

This single-center, exploratory, prospective and descriptive clinical study (ClinicalTrials.gov: NCT04398394; kofam.ch: SNCTP000003921) was performed in the Clinical Investigation Center of the Jules-Gonin Eye University Hospital in Lausanne.

Eligibility criteria and recruitment: Patients with a clinical diagnosis of dry AMD were enrolled in the study between August 2020 and April 2022. Patients were eligible for inclusion in the study if they were over 50 years of age, presented with visual acuity of at least 0.6 and had a clinical judgment of good central fixation. Clinical staging of dry AMD was performed by retinal specialists using the standardized classification proposed by the Beckman Initiative for Macular Research Classification Committee [26]. This system classifies AMD into early, intermediate and advanced (with geographic atrophy) stages based on fundus characteristics such as drusen size, number and the presence of pigmentary abnormalities.

Exclusion criteria included eyes with significant anterior segment opacities, foveal geographic atrophy, neovascular stage of AMD, pigment epithelium detachment, unclear clinical situation (such as unconfirmed retinal diagnosis or multiple pathologies) and eyes with high myopia (<−6D), high hypermetropia (>+5D) and/or high astigmatism (>+4 diopters). Patients in the following clinical situations were excluded from participation: having less than three months post-surgery of the anterior segment (e.g., cataracts), less than six months post-surgery of the posterior segment (e.g., vitreoretinal surgery), or with active uveitis. Furthermore, patients with albinism, epilepsy, inability to follow study procedures, inability to fix a target for at least 20 s, or a lack of tolerance for being in the dark for 30 min were excluded from the study. Participants were free to depart from the study at any time without penalty. All participant recruitment was conducted in the Jules-Gonin Eye Hospital in the medical retina service. After providing informed consent, patients underwent a screening process and an AO-TFI examination as part of the baseline examination, both of which are described below.

Baseline and screening visit: At the initial baseline visit, screening measures and a standard ophthalmological examination yielding best-corrected visual acuity (BCVA), intraocular pressure (IOP) (Icare IC100 TA011, Medilas AG/I care, Helsinki, Finland), spherical equivalent refractive error (RE) (NIDEK RT-6100, NIDEK CO, Tokyo, Japan) and axial length (AL) (IOL MASTER 700, Carl Zeiss Meditec AG, Jena, Germany) were performed. Demographic information (i.e., sex, age, etc.) and medical history were also collected at this time. On the same day, a variety of retinal images were collected using different methodologies. These included spectral-domain OCT (Spectralis camera, Heidelberg Engineering, Inc, Franklin, MA, USA), color and FAF photography (Optos ultra-widefield camera, Nikon, Tokyo, Japan). For each eye, 5–6 high-resolution images were acquired using AO-TFI imaging modality with the first-generation Cellularis^®^ camera (EarlySight SA, Geneva, Switzerland). The acquisition time for these images was 30 min, and the total AO-TFI examination process lasted 60 min.

Acquisitions of AO-TFI images: AO-TFI imaging was performed at the screening visit by trained operators, including two physician research assistants and two optometrists using the retinal camera prototype for AO-TFI. The light of two IR LED beams transmitted through the sclera provided the oblique illumination of the posterior retina, which was then imaged using a trans-pupillary AO full-field camera system. The details have been previously described elsewhere [17,18]. This technique provides an image with a field depth of 60 µm, and a lateral resolution of 3 µm if the pupil is dilated to 6 mm or more. The patients’ head and eye position were aligned with the system using a chin rest. For each acquisition, 100 raw images were captured in 8 milliseconds (20 s for single-layer imaging) and then post-processed to produce a single high signal-to-noise ratio image. In-vivo images of RPE cells were arbitrarily acquired in 6 zones (Z): 4 around the fovea at an eccentricity of 3.8° (Z1 to Z4), 1 foveal (Z5) and 1 at the discretion of the investigator (Z6). For each acquisition, the software recorded the iris image to check the alignment of the eye and a low-resolution oblique-illuminated 30° × 30° IR fundus to locate the high-resolution RPE images.

AO-TFI image correlations with OCT B-scans: AO-TFI image quality was first assessed qualitatively by a retinal specialist. For a region to be considered gradable and included in the correlative analysis, both image structure contrast and noise level were evaluated. Additionally, a quantitative assessment was performed using the beta version of a proprietary software (EarlySight SA, v. 1). This software applies a deep-learning framework to evaluate image quality based on parameters such as focus, noise level and the presence of RPE mosaic. The resulting quality factor ranges from 0 to 1, with the following categories: poor (<0.1), moderate (0.1–0.3), good (0.3–0.6) and excellent (≥0.6). The method used to correlate the mosaics of the AO-TFI images with the IR fundus images and the OCT B-scan was performed with a dedicated Fiji software tool (ImageJ, version 1.53q) as previously described [19]. Briefly, mosaics were generated by stitching together AO-TFI images acquired from each eye using the MosaicJ/TurboReg [27] Fiji plugins, and the resulting montage was analyzed using a semi-automated method. This analysis involved the registration of the mosaics with IR fundus images using the Bigwarp plugin [28], and then the correlation of the warped AO-TFI mosaics with IR fundus and their corresponding OCT B-scans using a custom plugin.

## 3. Results

The present study included 11 eyes from 9 patients, 8 of whom were female and 1 male, with a mean age of 70 years ± 8.3 years (range, 56–80 years). The baseline characteristics of the included eyes are provided in Table 1.

The AMD grades in the selected eyes were as follows: (1) early AMD in five eyes, one with a transparent lens and four with non-significant cataracts; (2) intermediate AMD in one eye with non-significant cataracts; (3) extrafoveal atrophic AMD in five eyes, one with non-significant cataracts and four pseudophakic (Table 2).

Following the AO-TFI image quality assessment, a total of 60 AO-TFI images were selected from the 66 predefined areas imaged in the 11 eyes. The mean quality of these images was found to be moderate (quality factor = 0.18 ± 0.09). After stitching the AO-TFI images from each eye into mosaics, these montages were meticulously correlated with the IR fundus and OCT images. This process illustrated in Appendix A enabled the identification of the AO-TFI patterns associated with the diverse pathological changes observed in dry AMD. Of particular interest were the following zones: (1) clinically healthy areas, (2) drusen, (3) subretinal drusenoid deposits (SDD) and (4) atrophic zones and borders.

### 3.1. Clinically Healthy Areas

OCT B-sections revealed that 24 out of the 60 imaged areas (40%) were deemed to be clinically healthy, 23 of these were observed in the 6 eyes diagnosed with early and intermediate AMD, while the remaining 1 was identified in an atrophic eye. The AO-TFI distinguished RPE cells in 14 images (58%) obtained from clinically healthy areas in eyes with early and intermediate AMD, though with varying contrast quality. A clear distinction of the classical RPE pattern, characterized by polygonal cells with hyporeflective centers and bright edges, was gradable in five AO-TFI images acquired from two eyes with early AMD (Figure 1b–e, blue insets). In the remaining images, individual cells were discernible but appeared discontinuous and blurred (Figure 1e,f, yellow rectangle).

### 3.2. Drusen of Various Size

A total of 78 drusen were observed in OCT sections, corresponding to 23 areas imaged in six eyes with early and intermediate AMD, as well as in one atrophic eye. Of these drusen, 49 (63%) were identified as cuticular drusen in OCT sections (Figure 2e,f, blue arrows) and characterized in AO-TFI images as structures with hyporeflective centers and bright edges (Figure 2d). A significant proportion of these structures corresponded to drusen of exceptionally small size on OCT, with the smallest being undetectable on OCT or manifesting as minor nodules of RPE thickening that could not be definitively identified (Figure 3d–f, blue inset). In the seven eyes that exhibited cuticular drusen, AO-TFI revealed 80 hyporeflective dots with bright edges, indicating 31 additional cuticular drusen-like structures compared to those observed in OCT sections. AO-TFI was determined to be the most sensitive imaging modality for detecting small cuticular drusen.

In contrast, larger drusen identified as soft drusen were not clearly characterized in AO-TFI images. While OCT identified approximately 30 soft drusen in 11 imaged areas, AO-TFI displayed ill-defined isoreflective regions, sometimes with a clear border, that were difficult to distinguish against the confounding black-and-white background pattern on AO-TFI images (Figure 3d,e, yellow inset).

### 3.3. Subretinal Drusenoid Deposits

Conversely, identification of early subretinal drusenoid deposits (SDD, stages 1 and 2) was challenging due to background interference in the AO-TFI images (Appendix A). However, when OCT revealed the presence of subretinal material with a conical shape disrupting the ellipsoid zone, a characteristic feature of stage 3 SDD [29] AO-TFI images revealed distinctive hyperreflective at the level of advanced SDD (Figure 4d–g, blue arrows). These findings were consistently observed in all nine regions imaged with AO-TFI, where OCT B-scans confirmed the presence of stage 3 SDD in three eyes with early AMD and two with atrophy.

### 3.4. Atrophic Regions and Borders

Nineteen AO-TFI images were acquired in extrafoveal atrophy across five eyes. In three eyes, choroidal vessels were not visible on either the color fundus or AO-TFI images (Figure 4c,d). In one eye, choroidal vessels, which appeared red on the color fundus image, were visible as hyporeflective vascular structures on AO-TFI images (Figure 5c,d, *). In the last eye, red and yellowish choroid in the fundus image appeared hyperreflective on AO-TFI images. (Figure 6c,d, *). In this latter case, OCT imaging revealed the absence of choriocapillaris, choroidal vessels with highly reflective walls where the fundus showed red and yellowish vessels (Figure 6f, *) and a white structure with a black border at locations where the fundus showed yellowish structures and AO-TFI displayed highly reflective vessels, suggesting the presence of nerves (Figure 6g, circle).

Within atrophic regions, the typical black-and-white background pattern observed on AO-TFI became isoreflective. Highly reflective crystalline structures were identified in areas where OCT sections revealed RPE and outer retinal atrophy (RORA; Figure 4d,e and Figure 5d,e, white circle). Conversely, hyporeflective dots, smaller than drusen and lacking a white ring, corresponded to regions with severely disturbed RPE over residual Bruch’s membrane and intraretinal hyperreflective foci seen on OCT (Figure 5d,e, arrows).

At the atrophic borders, where the double-layer sign was visible in OCT B-scans, the AO-TFI pattern remained isoreflective, making it difficult to delineate the atrophy boundaries (Figure 4d–f and Figure 5d–f). In these areas, numerous dark sports were identified on AO-TFI, corresponding to pigmented residual cells visible on color fundus images (Figure 4c,d, yellow circle). In the absence of RPE displacement associated with the thin double-layer sign, high contrast zones with a pronounced yet irregular black-and-white pattern were observed in AO-TFI images, allowing them to delineate the atrophy boundaries (Figure 6d). These regions corresponded to areas in OCT sections where the RPE appeared disrupted but not yet atrophic (Figure 6e).

## 4. Discussion

This observational study aimed to describe images of the first-generation AO-TFI system to visualize RPE layer in patients with non-exudative AMD. A total of 60 AO-TFI images from 11 eyes of 9 patients at different AMD stages were analyzed. While OCT provides high-resolution cross-sectional imaging of retinal layers, en face imaging techniques such as AO-TFI offer a broader field of view and a global perspective of RPE organization. We have previously demonstrated that normal RPE cells can be visualized using AO-TFI, allowing morphologic and quantitative comparisons with RPE flat mounts [18]. In the context of AMD, OCT imaging has revealed a wide spectrum of changes at the RPE level, including different types of drusen, subretinal and sub-RPE deposits, RPE elevations, RPE drusenoid detachments, vitelliform lesions, hyperreflective foci and RPE loss, reflecting the substantial and diverse remodeling that occurs over the course of the disease and even within a single eye [30]. These lesions may predict susceptibility to geographic atrophy progression. The aim of our study was to identify specific AO-TFI patterns associated with these AMD-related changes and correlate them with SD-OCT and fundus imaging.

AO-TFI features in dry AMD: from early to atrophic stages: AO-TFI has successfully visualized RPE cells in some clinically healthy areas, particularly in eyes with early AMD. However, in most cases, the visibility of normal RPE cells appears to be significantly compromised. This may be attributed to aging of the ocular media, which increases optical aberrations, and/or early alterations in pigment metabolism associated with AMD development [6,31,32]. Given this limitation, the current study focused on qualitative characterization of AO-TFI features and did not aim to compare RPE cell metrics across disease stages or with healthy controls.

The patterns on AO-TFI have been found to differentiate between different types of drusen. Notably, AO-TFI demonstrated superior sensitivity in detecting small cuticular drusen, which appeared as hyporeflective spots with bright edges, even when they were barely visible on SD-OCT. This suggests that AO-TFI may enhance early AMD characterization. In contrast, larger soft drusen were not clearly distinguishable due to interference from the background pattern. While advanced SDD appeared as distinct hyperreflective spots on AO-TFI, early stage SDD was more challenging to identify, likely because RPE cell organization and pigment distribution remain largely preserved at this stage.

Atrophic regions exhibited characteristic AO-TFI patterns. The typical black-and-white background pattern became isoreflective. Highly reflective crystalline structures were identified in areas with RORA, while hyporeflective dots were observed in areas with severe RPE disruption. Thus, AO-TFI can effectively differentiate pigment-rich—hyporeflective in AO-TFI but hyperreflective in OCT—from cellular debris containing photoreceptor segments in RORA, which appear hyperreflective in both imaging modalities. Atrophic borders were generally isoreflective, making delineation difficult; however, in the absence of RPE displacement, high-contrast zones with a characteristic black-and-white pattern were observed in pre-atrophic regions, allowing more precise boundary definition.

Comparison with other imaging modalities: AO-SLO and AO-FI has been used for high-resolution imaging of AMD-related changes, offering detailed visualization of retinal structures [20,21,22,23,24,25]. These studies have reported abnormal RPE morphology [20,22], loss of individual cell delineation [23] and identification of drusen [21,24,25], while also identifying novel features such as punctate hyperreflectivity and mobile melanin-containing clumps [33]. These features may play a role in AMD progression. Despite its advantages, AO-SLO has limitations, including a restricted field of view, sensitivity to eye motion necessitating patients to maintain steady eye fixation over an extended period and long acquisition times [17,24,34]. Additionally, it is also important to note that while the use of AO modalities has been studied in healthy volunteers and patients with various retinal diseases, research in AMD patients remains an active area that has not been extensively explored. This is due to the fact that all AO-assisted imaging requires clear optics, which can be challenging in elderly patients with early cataracts.

Compared to transpupillary illumination-based methods, AO-TFI enhances visualization of deeper retinal layers, particularly the RPE. The results of our study indicate that AO-TFI is highly sensitive for detecting small drusen and early RPE alterations, potentially offering a complementary imaging tool for AMD assessment. However, unlike AO-SLO, the AO-TFI system used in this study does not provide high-resolution imaging of photoreceptors.

Clinical implications and future directions. Our findings suggest that AO-TFI may serve as a valuable tool for detecting early structural changes in AMD, particularly small drusen and pre-atrophic RPE alterations. The ability to visualize characteristic high-contrast patterns in pre-atrophic regions raises the possibility that these patterns may serve as early markers of RPE changes, preceding the full manifestation of a mature atrophic state. However, further studies are needed to elucidate the mechanisms underlying these patterns. Beyond its diagnostic capabilities, the ability of AO-TFI to detect early cellular pathology may have implications not only for the pathogenesis of AMD but also for the development of emerging treatments [35]. Early detection of RPE changes may facilitate monitoring of novel therapeutic approaches, including complement cascade inhibitors targeting inflammation [36], photobiomodulation as a non-invasive low-level laser therapy [37] and RPE cell transplantation [38]. AO-TFI may play a key role in assessing treatment response by providing high-resolution imaging of subtle structural changes in the retina. This study represents the first clinical application of AO-TFI in dry AMD.

While the findings are promising, the study’s limitations include a small sample size (11 eyes of 9 patients), the absence of longitudinal data and the lack of correlation with functional measures. Consequently, the conclusions of this first-in-human study of AO-TFI in non-exudative AMD remain exploratory and will serve as a foundation for subsequent studies. Future research should focus on larger patient cohorts and longitudinal studies to assess the diagnostic precision and efficacy of this modality. The next-generation AO-TFI system with transpupillary illumination for simultaneous high-resolution imaging of photoreceptors and RPE represents a major step forward. It may help overcome current limitations.

## 5. Conclusions

AO-TFI is a promising imaging modality that provides complementary information in non-exudative AMD. Its high sensitivity for detecting small cuticular drusen and early RPE changes suggests potential utility in early AMD characterization. While it does not replace OCT or AO-SLO, its ability to visualize en face RPE changes makes it a valuable addition to the imaging armamentarium for AMD research. Future advances in AO-TFI technology may further improve its diagnostic and monitoring capabilities, paving the way for earlier and more precise AMD management.

## Figures and Tables

**Figure 1 cells-14-00633-f001:**
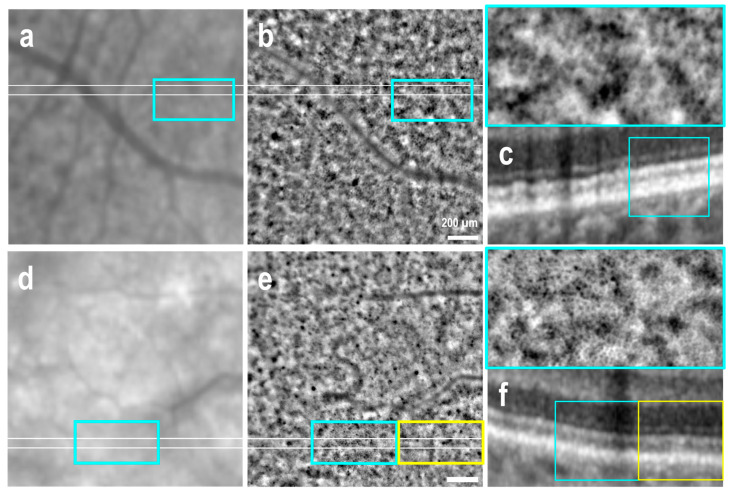
Retinal pigment epithelial cells in healthy areas of two eyes with early AMD. IR fundus images (**a**,**d**) correspond to the AO-TFI images (**b**,**e**). The white lines indicate the locations of the OCT B-sections (**c**,**f**). In areas where the IR images were isoreflective and the OCT B-scans showed no RPE changes, the AO-TFI images revealed the normal RPE pattern, characterized by polygonal cells with hyporeflective centers and bright edges, arranged in a honeycomb pattern in a few areas (**b**,**e**, blue insets). In the remaining areas, individual cells were visible but appeared blurred (**e**, yellow rectangle). (**a**–**c**: left eye, female, 72 years) Images extracted from stack image 103 of the full “Correlation fundus-AO-TFI-OCT” dataset available in Appendix A. (**d**–**f**: right eye, male, 63 years) Images extracted from stack image 15 of the full “Correlation fundus-AO-TFI-OCT” dataset available in Appendix A.

**Figure 2 cells-14-00633-f002:**
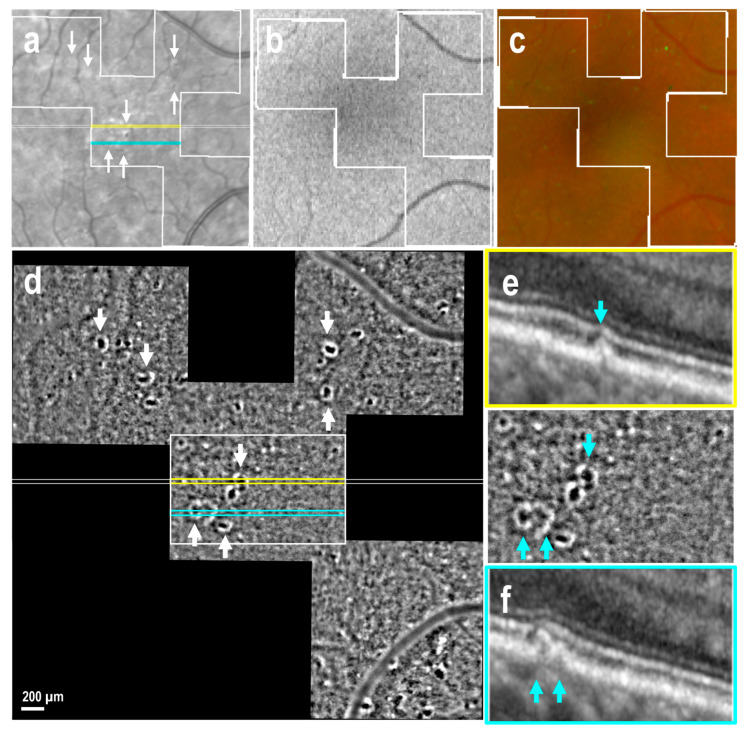
Cuticular drusen in early AMD (right eye, female, 67 years). IR fundus (**a**), FAF (**b**) and color fundus (**c**) images show the mask (white lines) of the area imaged with AO-TFI. The colored lines on the IR image indicate the locations of the OCT B-scans (**e**, yellow line; **f**, blue line). The AO-TFI mosaic (**d**) showed numerous structures with hyporeflective centers and bright edges. Some of these features were visible on IR image at much lower resolution (**a**, white arrows), but not on the FAF and color images. OCT B-scans showed that these features correspond to cuticular drusen (**e**,**f**, blue arrows). (**a**,**d**–**f**) Images extracted from stack images 57 and 66 of the full “Correlation fundus-AO-TFI-OCT” dataset available in Appendix A.

**Figure 3 cells-14-00633-f003:**
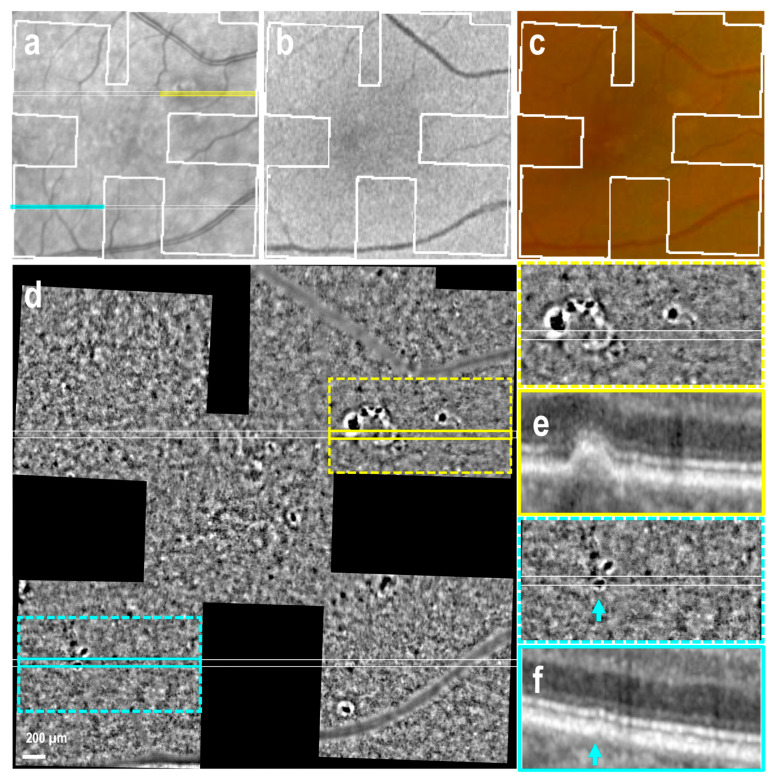
Cuticular drusen-like features and soft drusen in intermediate AMD (right eye, female, 62 years). IR fundus (**a**), FAF (**b**) and color fundus (**c**) images show the mask (white lines) of the area imaged with AO-TFI. The colored lines on the IR image indicate the locations of the OCT B-scans (**e**, yellow line; **f**, blue line). The AO-TFI mosaic showed “cuticular drusen-like” features, i.e., structure with hyporeflective centers and bright edges (**d**, blue inset), that were not detectable on OCT B-scan (**f**, blue arrow). Soft drusen identified on OCT B-scan (**e**) appeared as poorly defined isoreflective regions, in this case with a clear border, on AO-TFI (**d**, yellow inset). (**a**,**d**–**f**). Images extracted from stack images 27 and 85 of the full “Correlation fundus-AO-TFI-OCT” dataset available in Appendix A.

**Figure 4 cells-14-00633-f004:**
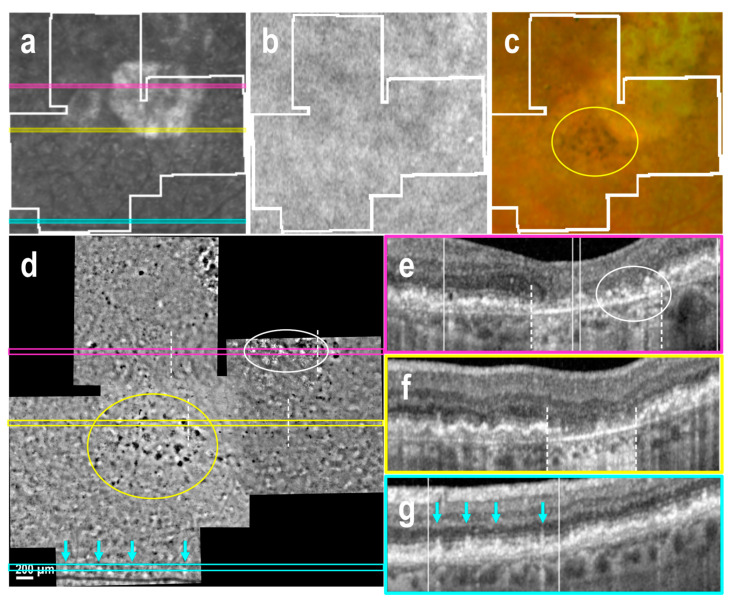
Atrophic AMD with double-layer sign, pigmented residual cells at the atrophic border and advanced subretinal drusenoid deposits (left eye female, 72 years). IR fundus (**a**), FAF (**b**) and color fundus (**c**) images show the mask (white lines) of the area imaged with AO-TFI. The colored lines on the IR image indicate the locations of the OCT B-scans (**e**, pink line; **f**, yellow line; **g**, blue line). In the atrophic region, which appeared hyperreflective in the IR fundus image (**a**) and yellow in the color fundus image (**c**), AO-TFI showed an isoreflective background with highly hyperreflective crystalline structures (**d**, white circle), corresponding to areas of RPE and outer retinal atrophy in OCT (**e**, white circle). At the atrophic border (dotted lines), where the double-layer sign was visible in OCT B-scans (**e**,**f**), the AO-TFI pattern remained isoreflective, while revealing pigmented residual cells corresponding to those seen in the color fundus image (**c**,**d**, yellow circle). In the inferior region, AO-TFI revealed prominent hyperreflective spots (**d**, blue arrow) at locations where the OCT scan showed advanced stage 3 SDD disrupting the ellipsoid zone (**g**, blue arrow). (**a**,**d**–**g**). Images extracted from stack images 7, 52 and 74 of the full “Correlation fundus-AO-TFI-OCT” dataset available in Appendix A.

**Figure 5 cells-14-00633-f005:**
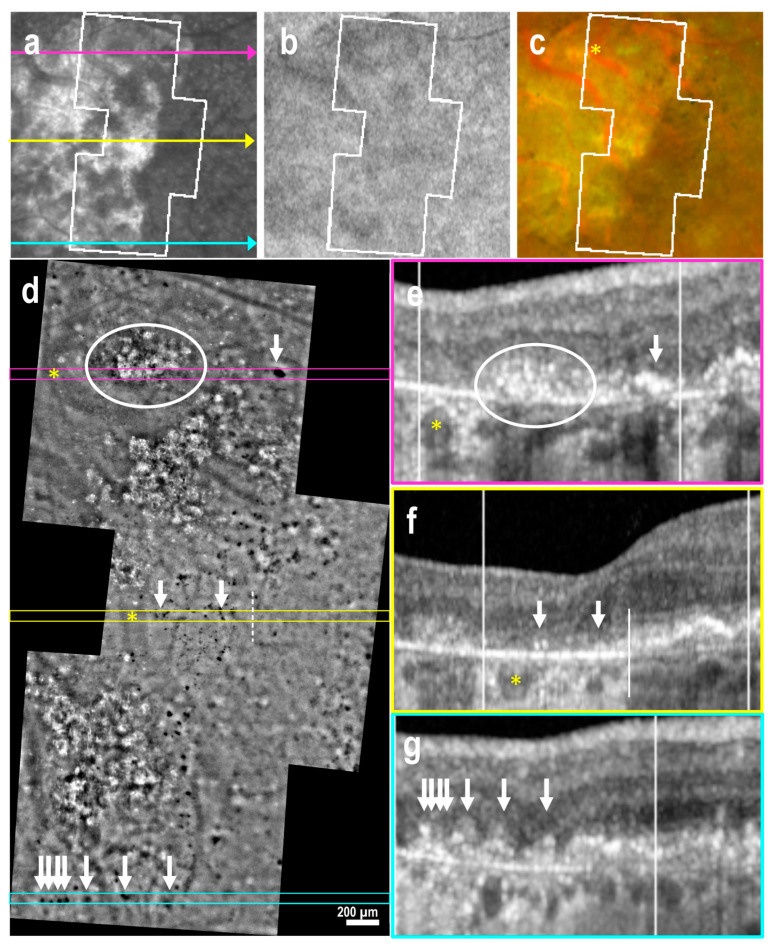
Geographic atrophy with double-layer sign at the border (right eye, female, 72 years). IR fundus (**a**), FAF (**b**) and color fundus (**c**) images show the mask (white lines) of the area imaged with AO-TFI. The colored lines on the IR image indicate the locations of the OCT B-scans (**e**, pink line; **f**, yellow line; **g**, blue line). In the atrophic region, which appeared hyperreflective in the IR fundus image (**a**) and yellow with red choroidal vessels in the color fundus image (**c**), AO-TFI showed an isoreflective background with hyporeflective choroidal vessels (**d**, *), highly hyperreflective crystalline structures corresponding to areas of RPE and outer retinal atrophy (RORA) in OCT (**d**,**e**, white circle) and dark dots corresponding to severely disturbed RPE in OCT (**d**–**g**, arrows). At the atrophic border (dotted lines), where the double-layer sign was visible in OCT B-scans (**f**), the AO-TFI pattern remained isoreflective (**d**). (**a**,**d**–**g**). Images extracted from stack images 7, 57 and 101 of the full “Correlation fundus-AO-TFI-OCT” dataset available in Appendix A.

**Figure 6 cells-14-00633-f006:**
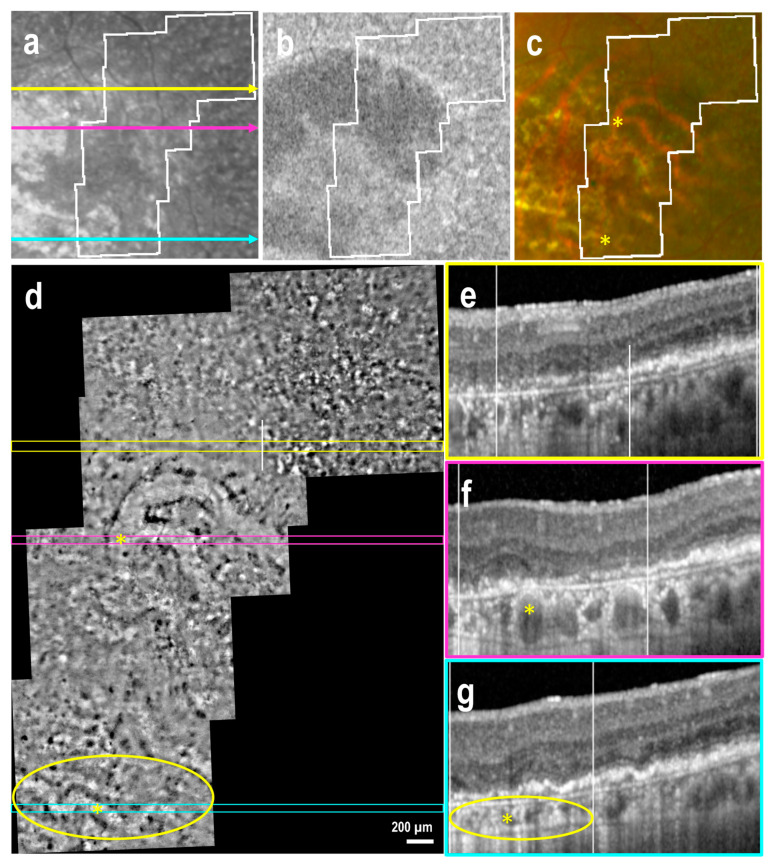
Geographic atrophy without double-layer sign at the border, and hyperreflective choroidal vessels (left eye, female, 80 years). IR fundus (**a**), FAF (**b**) and color fundus (**c**) images show the mask (white lines) of the area imaged with AO-TFI. The colored lines on the IR image indicate the locations of the OCT B-scans (**e**, yellow line; **f**, pink line; **g**, blue line). In the atrophic region, which appeared both hyperreflective and hyporeflective in the IR fundus image (**a**), hypo-fluorescent in the FAF image (**b**), and characterized by red and yellowish choroidal vessels in the color fundus image (**c**) and with severe RPE alteration in OCT (**e**–**g**), AO-TFI showed an isoreflective background with hyperreflective choroidal vessels (**d**, *). At the atrophic border (dotted lines), mild RPE alteration detached from Bruch’s membrane in the OCT B-scan (**f**, right of the dotted line) corresponding with a high contrast zone on AO-TFI (**d**). (**a**,**d**–**g**). Images extracted from stack images 10, 67 and 87 of the full “Correlation fundus-AO-TFI-OCT” dataset available in Appendix A.

**Table 1 cells-14-00633-t001:** Data and descriptive statistics of the participant and eye baseline characteristics.

Patient	Gender	Age (Years)	Eye	Iris Color	Axial Length (mm)	Spherical Equivalent RE (Diopters)	BCVA (Logmar)	IOP (mm Hg)
P041	F	67	OD	Blue	22.82	0.00	−0.097	14
P042	F	72	OD	Blue	24.16	−1.25	0.000	17
P042	F	72	OS	Blue	23.49	−0.50	0.000	15
P058	F	77	OD	Brown	22.99	0.75	0.000	17
P059	F	79	OD	Brown	24.07	−2.38	0.097	16
P066	F	80	OD	Brown	23.43	−5.75	0.000	13
P066	F	80	OS	Brown	23.54	−3.25	0.000	12
P072	F	56	OS	Brown	22.49	0.00	−0.097	12
P076	M	63	OD	Blue	26.25	−3.13	−0.097	12
P079	F	72	OS	Light brown	22.63	1.25	0.000	16
P085	F	62	OD	Blue	22.74	0.00	−0.097	15
	Mean	70			23.51	−1.30	−0.026	14.45
	SD	8.3			1.07	2.11	0.063	1.97
	Min	56			22.49	−5.75	−0.097	12
	Median	72			23.43	−0.5	0.000	15
	Max	80			26.25	1.25	0.097	17

**Table 2 cells-14-00633-t002:** AMD grade, lens status and number of 5° × 5° areas imaged using AO-TFI in the eyes included in the study. SD, standard deviation.

Patient	Eye	Diagnosis	Lens Status	N (Images)	Quality Mean	Quality SD
P041	OD	Early AMD	Non-significant Cataract	5	0.09	0.04
P058	OD	Early AMD	Non-significant Cataract	6	0.15	0.05
P072	OS	Early AMD	Transparent	4	0.22	0.05
P076	OD	Early AMD	Non-significant Cataract	6	0.39	0.03
P079	OS	Early AMD	Non-significant Cataract	6	0.28	0.07
P085	OD	Intermediate AMD	Non-significant Cataract	6	0.19	0.07
P042	OD	Atrophic AMD	Pseudophakia	4	0.13	0.03
P042	OS	Atrophic AMD	Pseudophakia	7	0.13	0.02
P059	OD	Atrophic AMD	Non-significant Cataract	6	0.17	0.03
P066	OD	Atrophic AMD	Pseudophakia	4	0.13	0.02
P066	OS	Atrophic AMD	Pseudophakia	6	0.13	0.03
			Total:	60	0.18	0.09

## Data Availability

Due to ethical restrictions, the data presented in this study are available on request from the clinical trial sponsor’s representative, Christophe Moser.

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
