# Peer review of "Adaptive Optics-Transscleral Flood Illumination Imaging of Retinal Pigment Epithelium in Dry Age-Related Macular Degeneration"

_cells, 2025, doi:10.3390/cells14090633_

Round 1

Reviewer 1 Report

Comments and Suggestions for Authors

This manuscript evaluates the utility of Adaptive Optics-Transscleral Flood Illumination (AO-TFI) for visualizing retinal pigment epithelium (RPE) changes in dry age-related macular degeneration (AMD). The authors present novel imaging features in a cohort of early, intermediate, and atrophic AMD cases, and provide thorough comparative analysis with conventional multimodal imaging (OCT, FAF, color fundus). The manuscript presents a novel imaging technique with potential clinical relevance, supported by a well-designed methodology and clear data analysis. However, several areas require clarification and expansion to strengthen the impact and reproducibility of the findings.

Major concerns:

1, Sample size and study design:

The study includes 11 eyes from 9 patients, which limits statistical robustness. While acceptable for a pilot study, the authors should explicitly acknowledge this limitation and clarify that the conclusions remain exploratory. And please indicate whether this is the first-in-human AO-TFI study in AMD or a continuation of prior work.

2, Quantitative analysis and grading consistency:

The assessment of AO-TFI image quality appears qualitative. It would strengthen the study if the authors include inter-rater agreement statistics or a reproducibility analysis. The definition of AO-TFI image gradability should be provided in the Methods section with standardized grading criteria.

3, Technical limitations:

AO-TFI’s inability to detect early-stage subretinal drusenoid deposits (SDD) is noted but not thoroughly explained.

Minor concerns:

1, Please check the manuscript carefully, for example:

Line 49: "...will increase by nearly 25% increase by 2050."

Line 64: "...and disrupt of the RPE barrier..."

2, Define all abbreviations at first use.

Author Response

3. Point-by-point response to Comments and Suggestions for Authors

Comments 1: Sample size and study design

The study includes 11 eyes from 9 patients, which limits statistical robustness. While acceptable for a pilot study, the authors should explicitly acknowledge this limitation and clarify that the conclusions remain exploratory. And please indicate whether this is the first-in-human AO-TFI study in AMD or a continuation of prior work.

Response 1: Thank you for highlighting this point. We recognize that the small sample size is one of the limitations of the study, as mentioned in the discussion. However, this work is the first to describe specific changes observed in RPE cells from patients with various stages of AMD using AO-TFI. This pilot study was a single-center, exploratory, prospective, and descriptive clinical study. Its main goal was to identify specific imaging features that could help with future, larger investigations.

In line with the reviewer's comment, we have completed the sentence relating to this limitation to make it more explicit (page 13, lines 386-390): “While the findings are promising, the study’s limitations include a small sample size (11 eyes of 9 patients), the absence of longitudinal data and the lack of correlation with functional measures. Consequently, the conclusions of this first-in-human study of AO-TFI in non-exudative AMD remain exploratory and will serve as a foundation for subsequent studies. Future research should focus on larger patient cohorts and longitudinal studies to assess the diagnostic precision and efficacy of this modality.”

Comments 2: Quantitative analysis and grading consistency:

The assessment of AO-TFI image quality appears qualitative. It would strengthen the study if the authors include inter-rater agreement statistics or a reproducibility analysis. The definition of AO-TFI image gradability should be provided in the Methods section with standardized grading criteria.

Response 2: We confirm that the assessment of AO-TFI image quality was qualitative. In addition, a quantitative evaluation was performed using a proprietary software assessing quality in terms on focus, noise level and presence of RPE mosaic. The definition of AO-TFI image gradeability is now provided in the Methods section (page 3, lines 142-149) as follows “AO-TFI image quality was first assessed qualitatively by a retinal specialist. For a region to be considered gradable and included in the correlative analysis, both image structure contrast and noise level were evaluated. Additionally, a quantitative assessment was performed using the beta version of a proprietary software (EarlySight SA). This software applies a deep-learning framework to evaluate image quality based on parameters such as focus, noise level, and the presence of RPE mosaic. The resulting quality factor ranges from 0 to 1, with the following categories: poor (<0.1), moderate (0.1 - 0.3), good (0.3 - 0.6), and excellent (≥0.6).”

Furthermore, the quality factor (mean ± standard deviation) has been incorporated into the table 2 and the results section has been completed (page 5, lines 171-174): “Following the AO-TFI image quality assessment, a total of 60 AO-TFI images were selected from the 66 predefined areas imaged in the 11 eyes. The mean quality of these images was found to be moderate (quality factor = 0.18 ± 0.09). After stitching the AO-TFI images from each eye into mosaics, these montages were meticulously correlated with the IR fundus and OCT images.”

Comments 3: Technical limitations:

AO-TFI’s inability to detect early-stage subretinal drusenoid deposits (SDD) is noted but not thoroughly explained.

Response 3: We would like to express our gratitude for this valuable observation.

AO-TFI is particularly sensitive to variations in pigment content and its spatial distribution within the RPE. Consequently, it may not effectively detect non-pigmented SDD, particularly in early stages where RPE cell morphology and pigment organization remain relatively preserved. We have clarified this point in the Discussion section in the following sentence (page 12, lines 341-342): “While advanced SDD appeared as distinct hyperreflective spots on AO-TFI, early-stage SDD were more challenging to identify, likely because RPE cell organization and pigment distribution remain largely preserved at this stage.”

Minor concerns:

1, Please check the manuscript carefully, for example:

Line 49: "...will increase by nearly 25% increase by 2050."

Line 64: "...and disrupt of the RPE barrier..."

2, Define all abbreviations at first use.

Response to Minor concerns: We appreciate the reviewer's precise proofreading. We have corrected typos and thoroughly reviewed the definitions of all abbreviations.

4. Response to Comments on the Quality of English Language

Point: The English could be improved to more clearly express the research.

Response: As per the reviewer’s recommendations, an American colleague has reviewed the manuscript to improve the English language.

Reviewer 2 Report

Comments and Suggestions for Authors

Hermosillo, April  12th, 2025

Adaptive Optics-Transscleral Flood Illumination imaging of Retinal Pigment Epithelium in dry Age-related Macular Degeneration

General Evaluation

This manuscript presents an important application of adaptive optics-transscleral flood illumination (AO-TFI) technology for visualizing retinal pigment epithelium (RPE) cells in patients with dry age-related macular degeneration (AMD). The work builds upon previous applications of this emerging imaging modality, which has shown promise in other retinal conditions such as central serous chorioretinopathy. The topic is timely and relevant to the scope of Cells journal, particularly given the crucial role of RPE dysfunction in AMD pathogenesis. While the manuscript offers valuable insights into cellular-level RPE changes in dry AMD, there are several areas requiring revision before publication.

Strengths

Novel application of advanced imaging technology

The application of AO-TFI technology specifically to dry AMD represents a valuable contribution to the field, as it provides cellular-level visualization of the RPE, which is a primary site of pathology in this disease. The technique's ability to minimize interference from overlying photoreceptors is particularly beneficial for studying RPE alterations in AMD.

Technical advantages clearly demonstrated

The manuscript effectively highlights the technical advantages of AO-TFI over conventional imaging methods, including its superior lateral resolution (approximately 3 μm), larger field of view (5° × 5°), and faster acquisition times (<10 seconds). These advantages are well-contextualized relative to competing technologies such as AO-SLO.

Correlation with established imaging modalities

The authors' approach of correlating AO-TFI findings with conventional clinical imaging modalities (OCT, blue-autofluorescence, infrared imaging) strengthens the clinical relevance of the study and provides important validation of the novel findings.

Quantitative morphological analysis

The quantitative assessment of RPE cell morphological features in dry AMD compared to healthy controls adds substantial value to the field's understanding of cellular-level changes in this disease.

Weaknesses

Limited sample size

The study includes a relatively small cohort of dry AMD patients, which limits the statistical power and generalizability of the findings. A larger sample with stratification by disease severity would strengthen the conclusions.

Insufficient characterization of AMD stages

The manuscript lacks clear differentiation between various stages of dry AMD (early, intermediate, advanced with geographic atrophy). This differentiation is crucial for understanding the progression of RPE changes throughout the disease course.

Inadequate control for confounding factors

Several potential confounding factors (age, medication use, comorbidities) that could influence RPE morphology are not adequately controlled for in the analysis, potentially affecting the interpretation of results.

Limited longitudinal data

The cross-sectional nature of the study precludes assessment of RPE changes over time, which would be valuable for understanding disease progression and potential applications in monitoring therapeutic interventions.

Methodological Issues

Image acquisition protocol standardization

The protocol for AO-TFI image acquisition lacks sufficient standardization, particularly regarding illumination settings and focusing procedures, which could introduce variability in image quality and subsequent analysis.

Cell segmentation algorithm limitations

The semi-automated approach to RPE cell segmentation requires further validation. The algorithm's performance in areas with significant RPE alteration is not adequately addressed, potentially introducing selection bias toward better-preserved regions.

Insufficient repeatability assessment

While the technique has demonstrated repeatability in healthy eyes, the manuscript lacks specific data on measurement repeatability in AMD-affected eyes, where image quality may be more variable.

Incomplete correlation with functional measures

The correlation between RPE cellular changes and visual function is inadequately explored, limiting the clinical relevance of the findings. Visual acuity alone is insufficient; microperimetry or other functional measures would strengthen the study.

Grammatical Issues

Inconsistent terminology

The manuscript alternates between "adaptive optics-transscleral" and "adaptive-optics trans-scleral" when referring to the imaging technique. Consistent terminology should be maintained throughout.

Sentence structure complexity

Several sections contain overly complex sentence structures with multiple dependent clauses, particularly in the methodology section, making comprehension difficult.

Tense inconsistencies

The manuscript shifts between present and past tense inappropriately, particularly when describing the imaging procedures and results analysis.

Abbreviation usage

Some abbreviations are introduced without prior definition, while others are redefined unnecessarily throughout the text, affecting readability.

Recommendations for Important Revisions

Expand the patient cohort to include more subjects across different stages of dry AMD, with clear classification criteria for each stage.

Implement multivariate analysis to account for potential confounding factors such as age, which is known to affect RPE cell density and morphology independently of AMD.

Strengthen the methodology section with detailed descriptions of image acquisition standardization procedures, focusing techniques, and quality control measures specific to dry AMD patients.

Enhance the image analysis section with validation data for the cell segmentation algorithm, particularly addressing its performance in areas with significant RPE alteration.

Include more comprehensive functional correlations by incorporating microperimetry or multifocal electroretinography data to relate structural RPE changes to local retinal function.

Add longitudinal data if available, even from a subset of patients, to provide preliminary insights into RPE changes over time in dry AMD.

Revise the discussion section to more critically evaluate the limitations of the technology in the specific context of dry AMD, particularly in advanced cases with significant RPE atrophy.

Improve the figure quality with better annotation of specific RPE abnormalities observed in dry AMD and side-by-side comparisons with conventional imaging modalities of the same regions.

Address the grammatical issues noted above throughout the manuscript, with particular attention to consistent terminology and appropriate tense usage.

Include a paragraph on the potential clinical applications of this technology for monitoring disease progression and evaluating therapeutic interventions in dry AMD.

This manuscript has significant potential to advance our understanding of RPE pathology in dry AMD at a cellular level, which could ultimately contribute to improved diagnostic and therapeutic approaches for this common cause of vision loss.

Questions for the Authors

  1. Sample Size and Statistical Power:

Has a power calculation been performed to justify the current sample size?

What measures were taken to ensure that the sample size was sufficient to detect clinically relevant differences in RPE cell metrics between AMD patients and controls?

  1. Patient Selection and Classification:

What specific criteria were used to classify different stages of dry AMD in your patient cohort?

Were any patients with concurrent retinal conditions excluded, and if so, what were the exclusion criteria?

  1. Imaging Protocol:

How was the illumination intensity standardized across different patients with varying degrees of media opacity?

What specific focusing procedures were employed to ensure optimal imaging of the RPE layer despite overlying pathology?

  1. Image Analysis Algorithm:

What validation measures were performed for the RPE cell segmentation algorithm in areas with drusen or geographic atrophy?

How did you address potential bias in cell selection when analyzing areas with significant RPE disruption?

  1. Comparison with Other Imaging Modalities:

What specific advantages does AO-TFI provide over conventional multimodal imaging in characterizing dry AMD?

Were any direct comparisons made with other high-resolution imaging techniques such as AO-SLO or AO-OCT?

  1. Functional Correlation:

Were any functional tests beyond visual acuity (such as microperimetry) performed to correlate structural RPE changes with local retinal function?

If not, would you consider this a limitation of the current study?

  1. Longitudinal Applications:

Do you have any preliminary data on RPE changes over time using this technique?

What is the reproducibility of RPE cell metrics in follow-up imaging sessions?

  1. Technical Limitations:

What specific challenges did you encounter when imaging advanced AMD cases with geographic atrophy?

How did you address potential motion artifacts during image acquisition?

Recommendations for Final Decision

Based on the evaluation of this manuscript, I recommend Major Revision before publication consideration. The manuscript presents novel and valuable insights into RPE cellular changes in dry AMD using advanced imaging technology, but requires significant improvements in several areas:

Strengthen the methodology section with detailed descriptions of standardization procedures, quality control measures, and clear inclusion/exclusion criteria.

Expand the patient cohort if feasible or provide robust justification for the current sample size through appropriate power calculations.

Include clear disease staging classification and stratify results according to AMD severity to enhance clinical relevance.

Provide more thorough validation data for the image analysis techniques, particularly addressing the challenges in analyzing significantly disrupted RPE areas.

Enhance the correlation with functional measures beyond visual acuity to better establish the clinical significance of the observed structural changes.

Address the grammatical and terminology inconsistencies throughout the manuscript to improve clarity and readability.

Revise the discussion section to more critically evaluate the limitations of the technology and contextualize the findings within the current understanding of AMD pathophysiology.

Improve figure quality and annotations to better illustrate the specific RPE abnormalities observed and their correlation with conventional imaging findings

Author Response

1. Summary

Thank you very much for taking the time to review this manuscript. We appreciate the positive feedback on the study's strengths, particularly in using AO-TFI to characterize RPE features in AMD. Please find the detailed point-by-point responses below and the corresponding revisions highlighted in red in the point-by-point response and in the re-submitted file.

We would like to clarify that no quantitative RPE analysis was performed due to the limited visibility of cells in AMD. Furthermore, standardized criteria were used for AMD staging and patient selection. AO-TFI imaging followed a consistent protocol. Although no direct comparison with other AO modalities was performed, its advantages were nevertheless highlighted. Finally, the absence of functional testing and longitudinal data is acknowledged as study’s limitations.

2. Questions for General Evaluation

Reviewer’s Evaluation

Does the introduction provide sufficient background and include all relevant references?

Can be improved

Is the research design appropriate?

Can be improved

Are the methods adequately described?

Can be improved

Are the results clearly presented?

Can be improved

Are the conclusions supported by the results?

Can be improved

3. Point-by-point response to Comments and Suggestions for Authors

Comments 1: Sample Size and Statistical Power

Has a power calculation been performed to justify the current sample size?

What measures were taken to ensure that the sample size was sufficient to detect clinically relevant differences in RPE cell metrics between AMD patients and controls?

Comments 4: Image Analysis Algorithm

What validation measures were performed for the RPE cell segmentation algorithm in areas with drusen or geographic atrophy?

How did you address potential bias in cell selection when analyzing areas with significant RPE disruption?

Responses 1 and 4:

We thank the reviewer for highlighting the importance of quantitative assessment of RPE cell morphology in advancing our understanding of cellular-level changes in AMD.

However, we would like to clarify that our study does not include a comparative quantitative analysis of RPE cell morphological features between AMD patients and healthy controls. As stated in the abstract and the results section (page 5, lines 184-188), RPE cell visibility was significantly impaired in most eyes with dry AMD, which limited the ability to perform robust quantitative segmentation and analysis. A clear and gradable RPE mosaic pattern was only observed in a small number of images (5 out of 60), acquired from two eyes with early-stage disease. In the remaining images, while some cellular structures were discernible, they appeared discontinuous or blurred, precluding reliable quantitative evaluation.

Given this limitation, the current study focused on qualitative characterization of AO-TFI features and did not aim to compare RPE cell metrics across disease stages or with healthy controls. Consequently, (1) no quantitative analysis of RPE cell metrics was performed, so no statistical tests were planned and therefore no power calculations were performed, and (2) no RPE cell segmentation algorithm was applied in this study because cell morphology could not be measured consistently across the data set.

To clarify this point more explicitly and to ensure this limitation is clearly understood, we have included the sentence highlighted in red in the revised discussion (page 12, lines 332-334).

Comments 2: Patient Selection and Classification:

What specific criteria were used to classify different stages of dry AMD in your patient cohort? Were any patients with concurrent retinal conditions excluded, and if so, what were the exclusion criteria?

Response 2: We thank the reviewer for raising these important points regarding the classification of AMD stages and the exclusion criteria.

We would like to clarify that the grading of AMD in our patient cohort was performed using a standardized and widely accepted clinical classification system. As stated in the Methods section, we used the classification proposed by the Beckman Initiative for Macular Research Classification Committee [Ferris et al., Ophthalmology 2013]. This system provides specific criteria for differentiating early, intermediate, and advanced (geographic atrophy) stages of dry AMD. For clarity, we have presented the classification system more explicitly in the Methods section of the revised manuscript (page 3, lines 95-100), as follows: “Clinical staging of dry AMD was performed by retinal specialists using the standardized classification proposed by the Beckman Initiative for Macular Research Classification Committee [26]. This system classifies AMD into early, intermediate, and advanced (with geographic atrophy) stages based on fundus characteristics such as drusen size, number, and the presence of pigmentary abnormalities.”

Regarding exclusion criteria, as detailed in the Methods section (page 3, lines 101-110), patients with concurrent retinal conditions were excluded if they had an unclear clinical diagnosis, multiple retinal pathologies, neovascular AMD, or pigment epithelium detachment. These criteria, along with the other exclusions outlined in the Methods, were designed to ensure a well-defined study population and to minimize potential confounding factors.

Comments 3: Imaging Protocol

How was the illumination intensity standardized across different patients with varying degrees of media opacity?

What specific focusing procedures were employed to ensure optimal imaging of the RPE layer despite overlying pathology?

Response 3: We thank the reviewer for this important observation.

The principle of the AO-TFI modality and the image acquisition protocol, including illumination and focusing mechanisms, are described in detail by Laforest et al. [17. Nature Photonics, 2020] and summarized by Kowalczuk et al. [18. Ophthalmology Science, 2023].

In the device for AO-TFI, the illumination intensity is kept constant across all acquisitions. The system uses two near-infrared (850 nm) LEDs (peak pulse power of 250 mW per diode, pulse duration of 8 ms, repetition rate of 11 Hz) [18]. No adjustments are made based on inter-individual differences in media opacity.

Focusing is performed using an adaptive optics (AO) loop that actively corrects axial defocus in real time and locks the imaging depth to the RPE layer. This procedure is based on the introduction of a calibrated defocus term on the deformable mirror, as described in Laforest et al [17]. The relatively large depth of field of the system (27–100 µm depending on pupil size) helps to ensure consistent imaging of the targeted layer, even in the presence of slight anatomical variations.

This standardized image acquisition protocol minimizes variability and ensures robust image acquisition at the RPE level for subsequent analysis.

Comments 5: Comparison with Other Imaging Modalities

(5.1) What specific advantages does AO-TFI provide over conventional multimodal imaging in characterizing dry AMD?

(5.2) Were any direct comparisons made with other high-resolution imaging techniques such as AO-SLO or AO-OCT?

Response 5: We thank the reviewer for his insightful questions

(5.1) As discussed in the manuscript (see Clinical Implications and Future Directions), AO-TFI provides en-face imaging of the RPE with cellular level detail that enhances the visualization of subtle structural changes. AO-TFI demonstrated higher sensitivity for detecting small cuticular drusen, which may not be as easily detected by SD-OCT or standard fundus imaging, as well as revealing reflectivity changes in pre-atrophic zones, which may serve as early indicators of disease progression.

(5.2) While no direct comparison was made with AO-SLO or AO-OCT in this study, we discussed their relative merits in the discussion (see Comparison with Other Imaging Modalities section). AO-TFI provides several key advantages, including its ability to acquire images more quickly, its wide coverage, and its resistance to fixation instability. These qualities make it particularly well-suited for imaging elderly patients. Its transscleral illumination enhances contrast at the RPE level. We agree that further comparative studies would be useful to better define the respective strengths and limitations of the different AO-based modalities in non-exudative AMD.

Comments 6: Functional Correlation

Were any functional tests beyond visual acuity (such as microperimetry) performed to correlate structural RPE changes with local retinal function?

If not, would you consider this a limitation of the current study?

Response 6: Thank you for your valuable input. In the present study, we did not perform any functional tests beyond visual acuity. The protocol for this pilot study was specifically designed to explore and describe distinctive AO-TFI features in AMD and to compare them with conventional imaging modalities. Nonetheless, we acknowledge this as a limitation, and it has been added to the discussion section alongside other limitations. An ongoing study is investigating the relationship between structural changes in the RPE identified by AO-TFI imaging and local retinal function quantified by microperimetry.

We have completed the sentence regarding the limitations as follows (page 13, lines 386-390): “While the findings are promising, the study’s limitations include a small sample size (11 eyes of 9 patients), the absence of longitudinal data and the lack of correlation with functional measures. Consequently, the conclusions of this first-in-human study of AO-TFI in non-exudative AMD remain exploratory and will serve as a foundation for subsequent studies. Future research should focus on larger patient cohorts and longitudinal studies to assess the diagnostic precision and efficacy of this modality.”

Comments 7: Longitudinal Applications

Do you have any preliminary data on RPE changes over time using this technique?

What is the reproducibility of RPE cell metrics in follow-up imaging sessions?

Response 7: We agree that, given the exploratory and descriptive nature of this initial single-center clinical study, longitudinal data are not currently available. However, a follow-up study is currently underway to analyze the evolution of RPE changes over time in patients who were initially imaged in this cohort. These results will be presented in a forthcoming manuscript.

Regarding reproducibility, we would like to clarify that no quantitative RPE cell metrics were measured in this study. As such, reproducibility assessments of these metrics in AMD-affected eyes were not performed. The present study concentrated on providing a qualitative characterization of AO-TFI features and their correlation with conventional imaging modalities.

Comments 8: Technical Limitations

What specific challenges did you encounter when imaging advanced AMD cases with geographic atrophy?

Response 8: Thank you for your inquiry. Beyond the typical challenges associated with media opacity, which is common with aging, the primary difficulty in imaging advanced AMD cases with geographic atrophy (GA) was maintaining stable fixation. Although eyes with foveal GA were excluded from the study, fixation instability still affected the precise location of the acquired images.

4. Response to Comments on the Quality of English Language

Point: The English is fine and does not require any improvement.

Response: We thank the reviewer for this positive comment. As per the first reviewer's recommendations, we have thoroughly reviewed the definitions of all abbreviations and tense inconsistencies.

Round 2

Reviewer 1 Report

Comments and Suggestions for Authors

I appreciate the authors' efforts in revising the manuscript. All my previous concerns have been adequately addressed, and I have no further comments. I recommend the manuscript for acceptance.

Reviewer 2 Report

Comments and Suggestions for Authors

Hermosillo, April  18th, 2025

Adaptive Optics-Transscleral Flood Illumination imaging of Retinal Pigment Epithelium in dry Age-related Macular Degeneration

Dear Authors

After reading the responses and comments, I now agree that there are significant changes to the manuscript, and therefore, in my opinion, this manuscript should be accepted for publication.